

# Extra high superoxide dismutase in host tissue is associated with improving bleaching resistance in "thermal adapted" and *Durusdinium trenchii*-associating coral

Jih-Terng Wang[1], Yi-Ting Wang[1], Chaolun Allen Chen[2], Pei-Jei Meng[3,4], Kwee Siong Tew[4,5], Pei-Wen Chiang[2] and Sen-Lin Tang[2]

[1] Department of Oceanography, National Sun Yat-Sen University, Kaohsiung, Taiwan
[2] Biodiversity Research Center, Academia Sinica, Taipei, Taiwan
[3] General Education Center, National Dong Hwa University, Hualien, Taiwan
[4] National Museum of Marine Biology and Aquarium, Pingtung, Taiwan
[5] Institute of Marine Biodiversity and Evolution, National Dong Hwa University, Pingtung, Taiwan

Corresponding authors
Jih-Terng Wang,
jtw100@mail.nsysu.edu.tw
Sen-Lin Tang,
sltang@gate.sinica.edu.tw

## ABSTRACT

Global warming threatens reef-building corals with large-scale bleaching events; therefore, it is important to discover potential adaptive capabilities for increasing their temperature resistance before it is too late. This study presents two coral species (*Platygyra verweyi* and *Isopora palifera*) surviving on a reef having regular hot water influxes via a nearby nuclear power plant that exhibited completely different bleaching susceptibilities to thermal stress, even though both species shared several so-called "winner" characteristics (*e.g.,* containing *Durusdinium trenchii*, thick tissue, *etc.*). During acute heating treatment, algal density did not decline in *P. verweyi* corals within three days of being directly transferred from 25 to 31 °C; however, the same treatment caused *I. palifera* to lose < 70% of its algal symbionts within 24 h. The most distinctive feature between the two coral species was an overwhelmingly higher constitutive superoxide dismutase (ca. 10-fold) and catalase (ca. 3-fold) in *P. verweyi* over *I. palifera*. Moreover, *P. verweyi* also contained significantly higher saturated and lower mono-unsaturated fatty acids, especially a long-chain saturated fatty acid (C22:0), than *I. palifera*, and was consistently associated with the symbiotic bacteria *Endozoicomonas*, which was not found in *I. palifera*. However, antibiotic treatment and inoculation tests did not support *Endozoicomonas* having a direct contribution to thermal resistance. This study highlights that, besides its association with a thermally tolerable algal symbiont, a high level of constitutive antioxidant enzymes in the coral host is crucial for coral survivorship in the more fluctuating and higher temperature environments.

## INTRODUCTION

Endosymbiosis between reef building corals and dinoflagellates assigned to the family Symbiodiniaceae (*LaJeunesse et al., 2018*) creates the foundation for one of the most diverse and productive marine ecosystems on the planet. However, this association is highly sensitive to rising seawater temperature. A rise of only 1∼2 °C above the summer average under moderate to high irradiance will likely disrupt symbiotic relationships by causing the expulsion of symbionts from the host, resulting in so-called "coral bleaching" (*Fitt et al., 2001*; *Lesser & Farrell, 2004*). Global warming enforces this impact by increasing the incidence of repeated beyond-threshold seawater temperatures, leading to more frequent coral bleaching events such as the devastating bleaching events that have occurred at the Great Barrier Reef in recent years (*Hughes et al., 2018*). Catastrophic impacts caused by global warming highlights the threat to coral survival, and thus evokes intensive attention on the mechanisms underpinning coral bleaching.

Despite coral bleaching having been intensively studied (reviewed in *Weis (2008)*, *Lesser (2011)* and *Van Oppen & Lough (2018)*), its mechanism(s) are still not well-understood. The main obstacles to uncovering these mechanisms are attributed to the high diversity in coral species and dynamic multiple partnerships among coral holobionts. Reef building corals are not only associated with intracellular photoautotrophs, but also a consortium of microorganisms including bacteria, archaea, viruses, fungi, and protists (*Van Oppen & Lough, 2018*). The coral holobiont functions as a unit to provide flexible stability when confronting environmental stressors. Therefore, it is necessary to approach coral bleaching from several aspects.

Accumulating data indicate that the main cause of coral bleaching might be attributed to the generation of reactive oxygen species (ROS) resulting from photoinhibition in Symbiodiniaceae algae and/or mitochondrial dysfunction in the coral host under thermal stress (*Cziesielski, Schmidt-Roach & Aranda, 2019*). Bursts of ROS damage essential biomolecules and signals and mediate cellular termination processes such as apoptosis, autophagy, and cell necrosis (*Weis, 2008*; *Lesser, 2011*). However, some biotic factors are capable of mitigating coral bleaching. For example, associations with the Symbiodiniaceae algae *Durusdinium trenchii* or other thermal resistant species might elevate coral bleaching thresholds (*Stat & Gates, 2011*; *Hsu et al., 2012*; *Howells et al., 2012*; *Silverstein, Correa & Baker, 2012*; *Keshavmurthy et al., 2014*). The comparatively high resistance of *Durusdinium* algae to heat stress might be attributed to a high level of fatty acid saturation in membranes (*Tchernov et al., 2004*). In addition to algal symbiont factors, the role of coral hosts in resisting bleaching is also emphasized (*Baird et al., 2009*). After an intensive survey following a mass bleaching, *Loya et al. (2001)* proposed that the winners might be the massive and thick-tissue corals. Several lines of evidence also suggest that coral hosts could protect their symbiosis from heat bleaching by regulating photo-protective and/or antioxidant systems (*Baird et al., 2009*; *Linan-Cabello et al., 2010*), increasing heterotrophy (*Grottoli, Rodrigues & Palardy, 2006*), or controlling symbiont cell densities (*Cunning et al., 2015*). Coral hosts under heat stress could also upregulate the expression of stress related genes (*e.g.,* heat shock proteins and ROS scavengers) (*Barshis et al., 2013*;

*Dixon et al., 2015*; *Kenkel, Meyer & Matz, 2013*; *Palumbi et al., 2014*) or express a more thermal resistant respiration enzyme (*e.g.*, malate dehydrogenase) (*Wang et al., 2019*).

Apart from the coral host and Symbiodiniaceae algae, growing evidence indicates that the coral-associating microbiome might contribute to coral health and resilience (reviewed by *Bourne, Morrow & Webster, 2016*). A notable member of the core coral microbiome is affiliated with the *Endozoicomonas* genus (*Gammaproteobacteria* class, *Hahellaceae* family), which can comprise as much as 90% of the microbiome and is associated with a wide range of coral species (*Neave et al., 2017* and the references therein). Ecological and genetic evidence also suggest that *Endozoicomonas* might benefit coral health and fitness (*Bourne et al., 2008*; *Neave et al., 2014*; *Ding et al., 2016*). The roles of *Endozoicomonas* on thermal bleaching susceptibility in corals are controversial. Despite several studies suggested that high abundance of *Endozoicomonas* were usually found in the non-bleached corals after bleaching events (*Bourne et al., 2008*; *Lee et al., 2015*; *Pantos et al., 2015*; *Glasl, Herndl & Frade, 2016*; *Ziegler et al., 2017*), some bleached corals were still associated with this microbial group (*Pogoreutz et al., 2018*; *Shiu et al., 2020*). Direct evidence to show the positive effect of associating microbiomes with coral bleaching was presented by inoculating a so-called probiotic microbiome consortium to *Pocillopora*, which partially mitigated coral bleaching from higher temperatures (*Rosado et al., 2019*). However, how long the coral will maintain the introduced bacteria and the mechanisms by which inoculated bacteria may mitigate coral bleaching are unclear and require further studying (*Blackall et al., 2020*).

To examine the roles of the coral host, algal symbionts, and associating bacteria, we applied herein an acute temperature treatment to test bleaching responses in two dominant coral species, *Platygyra verweyi* and *Isopora palifera*, in a natural mesocosm having seawater temperatures similar to forecasted 2050 ocean temperatures (*Hsu et al., 2012*; *Kao et al., 2018*). These species were selected due to their being *Durusdinium trenchii*-associated at our long-term monitoring site. This allowed us to compare the roles of coral hosts and their associated bacteria in their bleaching responses under the same algal symbiont to mimic the scenario of corals switching (or shuffling) to a thermal tolerant algal strain association following a thermal bleaching event.

## MATERIALS AND METHODS

### Sample collection, maintenance, and menthol bleaching

Nubbins of *P. verweyi* and *I. palifera* corals were collected from tagged populations (*Hsu et al., 2012*; *Kao et al., 2018*) at a 3 m depth. Pingtung County Government and National Kenting National Park approved this research. The sampling site was located in a bay near the third nuclear power plant outlet (NPP-OL) in Kenting National Park, Taiwan (120°44′13′E, 21°56′4′N), within a zone that has been heated by power plant discharges since 1985 (*Keshavmurthy et al., 2014*). Surviving populations *P. verweyi* and *I. palifera* corals at NPP-OL have adapted to the higher temperature environment by associating with the thermal resistant Symbiodiniaceae alga *Durusdinium trenchii* (*Hsu et al., 2012*; *Keshavmurthy et al., 2014*; *Kao et al., 2018*) since at least a 2007 survey. *Cladocopium* sp. C1 associating coral *Stylophora pistillata*, which has disappeared from the shallow waters

of NPP-OL (*Keshavmurthy et al., 2014*), was collected from below the hot water zone at 10–15 m depths at NPP-OL as the temperature sensitive reference. Collected coral nubbins were acclimated in a mesocosm aquarium as indicated in *Wang et al. (2012)* for one week to allow for wound healing prior to use.

Prior to determine the respiration breaking temperature (RBT) and fatty acid (FA) composition of the coral hosts, each of the approximately $3 \times 3$ cm coral nubbins were bleached with 90 ppm menthol-supplemented artificial seawater (ASW) prepared from a sea salt mixture (Instant Ocean, Aquarium Systems, France) as previously described (*Wang et al., 2012*).

## Tissue thickness, RBT values, and fatty acid composition analyses

Each of the approximately $2 \times 2$ cm *P. verweyi* and *I. palifera* nubbins was fixed in a 10% ASW-formalin solution, and then decalcified with 10% acetic acid containing 5% formalin for further tissue thickness measurement as described in *Loya et al. (2001)*.

Coral host RBT was measured following the method developed in *Wang et al. (2019)*. Basically, the changes in respiration rate of a menthol-bleached coral nubbins were monitored in a respiration chamber under 2 °C incrementally increasing temperatures from 26~27 °C to 42~43 °C. Oxygen content was measured with an optical dissolved oxygen instrument (YSI ProODO; YSI, Inc., Yellow Springs, OH, USA) to avoid electrode polarization.

To determine the FA composition of coral host lipids, menthol-bleached coral nubbins were starved for a week to reduce stored lipids before subjecting to lyophilization. Freeze-dried coral tissue was collected by cutting off the part above the basal plate with a bone saw. Coral tissue lipids were extracted with 10 volumes of chloroform/methanol (2:1, containing 0.01% 2,6-di-tert-butyl-4-methylphenol) at 4 °C overnight after powdering the tissue on the skeleton in liquid nitrogen with a mortar and pestle. Following dehydration, concentration by evaporation, and derivatization, methyl esterified FA was analyzed by gas chromatography. The degree of saturation of each FA was calculated from the ratio of percent saturated to all unsaturated derivatives.

## Heat treatment, fluorescence methodology, and sample preparation

Acute thermal challenging was used to compare the bleaching susceptibility of the two "thermal-adapted" coral species and a temperature-sensitive species. Heat treatment was conducted by transferring each of four coral nubbins (about $2 \times 2$ cm or five cm branch) from three colony replicates of one species, a total of 12 coral nubbins, from 25 °C aquarium water directly into to a glass tank with 40 L pre-warmed ($31.0 \pm 0.5$ °C) ASW. Seawater in the heated tank was filtered through several layers of aquarium filter pad and a protein skimmer. Before introducing the next species, the tank was cleaned and its ASW replaced with fresh ASW and circulated for one day. Temperature and light fluctuation in the heating tank were recorded with HOBO Data Loggers (UA-001-64). An example of temperature and light conditions is shown in Fig. S1, in which one or two sudden drops in light intensity during the daytime occurred during a 30 min dark adaptation in preparation for quantum yield measurements on the coral specimens.

During heating, the maximum quantum yield of PSII ($F_v/F_m = [F_m - F_0]/F_m$) of the coral nubbins were determined as described in *Wang et al. (2011)*. At each time interval, one nubbin from each colony replicate was collected for preparing tissue homogenates by air/buffer blasting with 50 mM phosphate buffer containing 0.1 mM EDTA and 10% glycerol (pH 7.0). After making up to 7 ml with the above buffer, tissue homogenates were centrifuged at 4 °C (15,000 rpm) for 15 min to collect pellets for algal density measurement and supernatants to determine stress enzyme activity.

Separate sets of coral nubbins from the same colony replicates of *P. verweyi* and *I. palifera* were also treated in the heated tank for 24 h, then washed twice with sterilized ASW and preserved in absolute ethanol at −20 °C for bacterial composition analysis.

## Algal density measurement and enzyme assay

The numbers of collected Symbiodiniaceae cells were counted with a Neubauer improved hemocytometer as previously described (*Wang et al., 2012*), and coral surface areas were normalized according to the aluminum foil method.

Total superoxide dismutase (SOD) activity in coral tissue extracts was determined with a commercial kit (19160 SOD determination kit, Sigma-Aldrich, USA) following manufacturer instructions. The catalase (CAT) assay was modified from the spectrophotometric method described by *Krueger et al. (2015)*. Briefly, 10 μl host extract was added to 790 μl 50 mM phosphate buffer containing 0.1 mM EDTA and 14 mM $H_2O_2$ (pH 7.0) for measuring changes at 240 nm absorbance, and finally converted to a $H_2O_2$ concentration having a molar coefficient of 43.6. All enzyme activity measurements were determined at 25 °C and standardized with the protein content in the extract as measured by a Bradford Assay (Bio-Rad Protein Assay; Bio-Rad, Hercules, CA, USA) according to the manufacturer's protocol.

## Bacterial composition analysis

Sequencing hypervariable regions V6-V8 of the 16S rRNA gene was used to examine bacterial composition in the corals. DNA extraction, PCR reaction, sequencing methodology, and data analysis followed the protocols described in *Shiu et al. (2017)*. Briefly, bacterial particles in treated coral tissue were isolated by air/buffer blasting with ×10 TE buffer and centrifugation (10,000× g, 15 min). DNA was extracted by a modified CTAB method, and then subjected to PCR amplification by using two bacterial universal primers targeting the V6-V8 hypervariable region of the 16S rRNA gene: 968 F (5′-AACGCGAAGAACCTTAC-3′) and Uni1391R (5′-ACGGGCGGTGWGTRC-3′). After agarose electrophoresis, PCR amplicons of approximately 420 bp were collected for further DNA tagging PCR amplification as developed by *Chen et al. (2011)*. The resulting DNA samples from tagging PCR were then applied to Illumina MiSeq sequencing (Yourgene Bioscience, Taipei, Taiwan), and the raw sequencing data merged into paired-end reads using USEARCH v8.1.1861 (*Edgar, 2010*).

To qualify sequence data, obtained reads were analyzed with Mothur v1.36.1 to retain the reads between 380 and 450 bp with Phred quality score > 27. Chimeric reads were then trimmed by UCHIME (*Edgar et al., 2011*) in USEARCH v7.0.1090 (parameters: reference

mode, rdp_gold database, and mindiv of 5). Resulting non-chimeric reads were further analyzed with UPARSE pipeline (*Edgar, 2013*) to generate OTUs (97% identity level). The OTU classification was performed on a per sample basis using the SILVA v128 database with a pseudo-bootstrap cut-off of 80%. Accession number to these SRA data is PRJNA736837: Bacterial diversity on Coral.

## Testing the contribution of *Endozoicomonas* to coral resistance to bleaching

To test the benefit of *Endozoicomonas* to coral thermal resistance to bleaching, *Endozoicomonas-* associated *P. verweyi* nubbins were treated with antibiotics and non-*Endozoicomonas-* associated *I. palifera* were infected with *E. montiporae* following 31 °C heat challenging. Antibiotic treatment was conducted by incubating *P. verweyi* in well-aerated ASW containing 100 $\mu$g ml$^{-1}$ nalidixic acid, 1 mg ml$^{-1}$ ampicillin, and 50 $\mu$g ml$^{-1}$ streptomycin for nine days at 25∼26 °C under the same light-dark maintenance regime mentioned above. Antibiotic-supplemented seawater was prepared with sterilized ASW and changed daily. One set of antibiotic-treated *P. verweyi* recovered in the maintenance aquarium (antibiotics free) for one week as the control. The infection of *I. palifera* with *Endozoicomonas* was modified from the method described in *Rosado et al. (2019)*. Basically, *Endozoicomonas* was cultured to the log phase in the medium developed by *Ding et al. (2016)*, and then concentrated by centrifugation at 12,000 rpm at 20 °C for 10 min, followed by adding ASW to make up a bacterial suspension containing $10^7 \sim 10^8$ cells ml$^{-1}$. To infect coral with *Endozoicomonas*, tested coral specimens were placed in a dish of ASW with only the coral skeleton immersed, leaving coral tissue exposed to the air. Then, 1 ml of concentrated bacterial suspension was spread onto coral tissues and allowed to stand for 10 min, followed by adding 1.5 L ASW into the dish for a further 2 h of incubation under aeration. Spreading corals with ASW was used as a control. The infection process was carried on for 2 h per day, and repeated for seven consecutive days. After infection, coral samples were rinsed with fresh ASW and transferred back to the maintenance aquarium. *Endozoicomonas* in coral tissues was detected by semi-quantitative PCR using mismatched primer En771R designed by *Shiu et al. (2018)* with β-actin gene as an internal reference. The primer set (F: TTCCTTGGAATGGAATCTGC; R: GCGAAGTGATTTCTTTCTGC) used to amply the β-actin gene was modified from *Shiu et al. (2020)*, and produces a 163 bp DNA fragment. Due to intensive algal symbiont loss occurring in *I. palifera* when treated with 31 °C 24 h, amplification of the bacterial 16S ribosomal RNA gene was conducted to confirm DNA extraction efficiency following the method described in *Chen et al. (2011)*. Heat challenging and monitoring the indicators of stress and bleaching were conducted as described above.

## Statistical analysis

Data in this study are presented as means $\pm$ S.D., with at least three colony replicates. Comparisons of enzyme activity between different incubation times were made using a one-way analysis of variance (ANOVA) followed by Tukey's *post hoc* analysis for multiple comparisons at a significance level of 0.05. The differences between two tested species

or between treatments within the same species were analyzed with a $t$-test. To reveal the differences in FA composition between the two tested species, a principal component analysis (PCA) was conducted. The similarity of γ-proteobacteria composition between the two coral species was analyzed with arc cosine-transformed relative abundance of the identified genus by multidimensional scaling (MDS) ordination (*Kruskal & Wish, 1978*), followed by an analysis of similarity (ANOSIM) (*Clarke & Warwick, 1994*) using PRIMER 6 (*Clarke & Warwick, 1994*).

## RESULTS

### Responses in photosynthesis and algal density to acute thermal stress

When coral nubbins were transferred directly from 25 to 31 °C, the maximum quantum yield of *P. verweyi* remained stable at > 0.610 throughout 72 h of heat treatment, but that of *I. palifera* bumped up and down between 0.514 and 0.606 between 8 and 32 h and then declined to about 0.340 after 48 h incubation (Fig. 1). Temperature-sensitive *S. pistillata*, as predicted, displayed a dramatic decrease in maximum quantum yield from about 0.660 to 0.170 within 24 h. Consistent with the performance in maximum quantum yield, the algal density of *P. verweyi* remained stable ($2.6\sim3.5 \times 10^6$ cells cm$^{-2}$) throughout 72 h of heat treatment. Conversely, the algal density of *I. palifera* and *S. pistillata* declined dramatically from $1.6 \pm 0.3 \times 10^6$ and $1.0 \pm 0.0 \times 10^6$ cells cm$^{-2}$ to $2.0 \pm 0.9 \times 10^5$ and $1.1 \pm 0.3 \times 10^5$ cells cm$^{-2}$ with the 24 h heat treatment, respectively. In summary, incubation at 31 °C caused the loss of $87 \pm 8\%$ and $89 \pm 3\%$ algal symbionts in *I. palifera* and *S. pistillata* within 24 h, respectively.

### Tissue thickness and physio-biochemical performance

To investigate why the two "thermal adapted" and *Durusdinium*-associated coral species, *P. verweyi* and *I. palifera*, displayed distinctly different bleaching susceptibility to acute thermal stress at 31 °C, several indictors known to highly correlate with coral bleaching were examined. Tissue thickness measurements indicated no difference between the two species ($P > 0.05$), which were $2.8 \pm 0.4$ mm for *P. verweyi* and $3.0 \pm 0.2$ mm for *I. palifera*. Respiration responses to rising temperature, expressed as RBT as described in *Wang et al. (2019)*, also showed no difference between two species ($P > 0.05$), which were $35.5 \pm 0.6$ °C for *P. verweyi* and $36.0 \pm 0.5$ °C for *I. palifera*.

FA composition in menthol-bleached and starved coral tissue of the two coral species indicated that, as shown in Fig. S2, more than 1/3 of the FA were dominated by myristic acid (14:0, *Platygyra*: $17.2 \pm 9.5\%$; *Isopora*: $15.9 \pm 4.3\%$) and palmitic acid (16:0, *Platygyra*: $17.0 \pm 5.2\%$; *Isopora*: $26.1 \pm 1.9\%$). *Platygyra* also contained one more dominant FA, behenic acid (22:0, $14.0 \pm 1.5\%$). The differences in FA composition between the two coral species were further examined by principal component analysis, in which >95% of the variance in the dataset could be accounted for by the first two principal components (PC) (Fig. 2A). As shown in Fig. 2A, the PC1, accounting for 62.1% of the variance, divided the data into two groups in which *P. verweyi* was majorly accounted for by containing a higher percentage of two saturated FA (behenic acid and myristic acid). In contrast, PC1

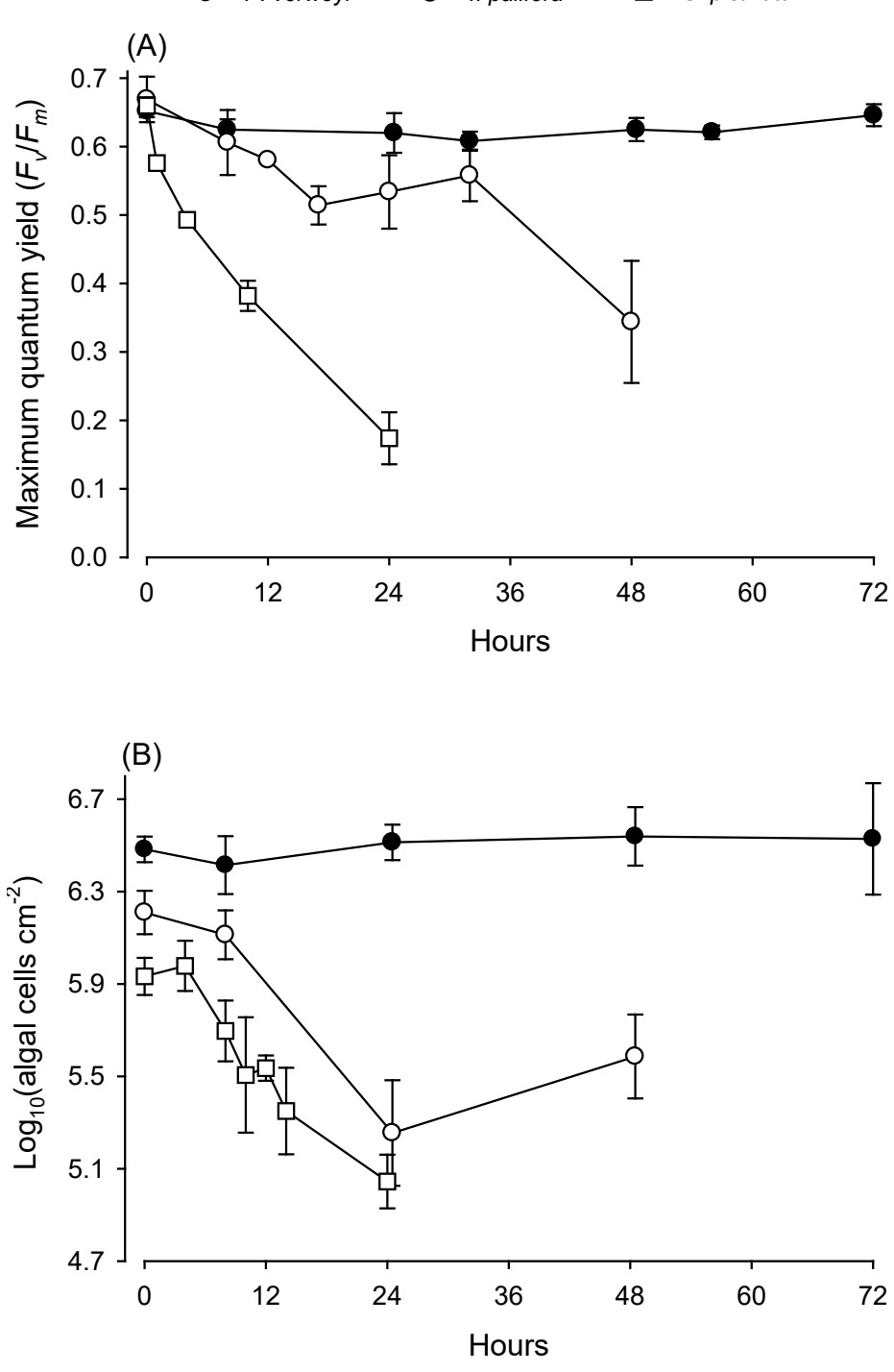

**Figure 1 Response of corals *Platygyra verweyi*, *Isopora palifera*, and *Stylophora pistillata* to acute heat stress at 31 °C.** Data points represent changes in maximum quantum yield of PSII (A) and algal density (B) over time.

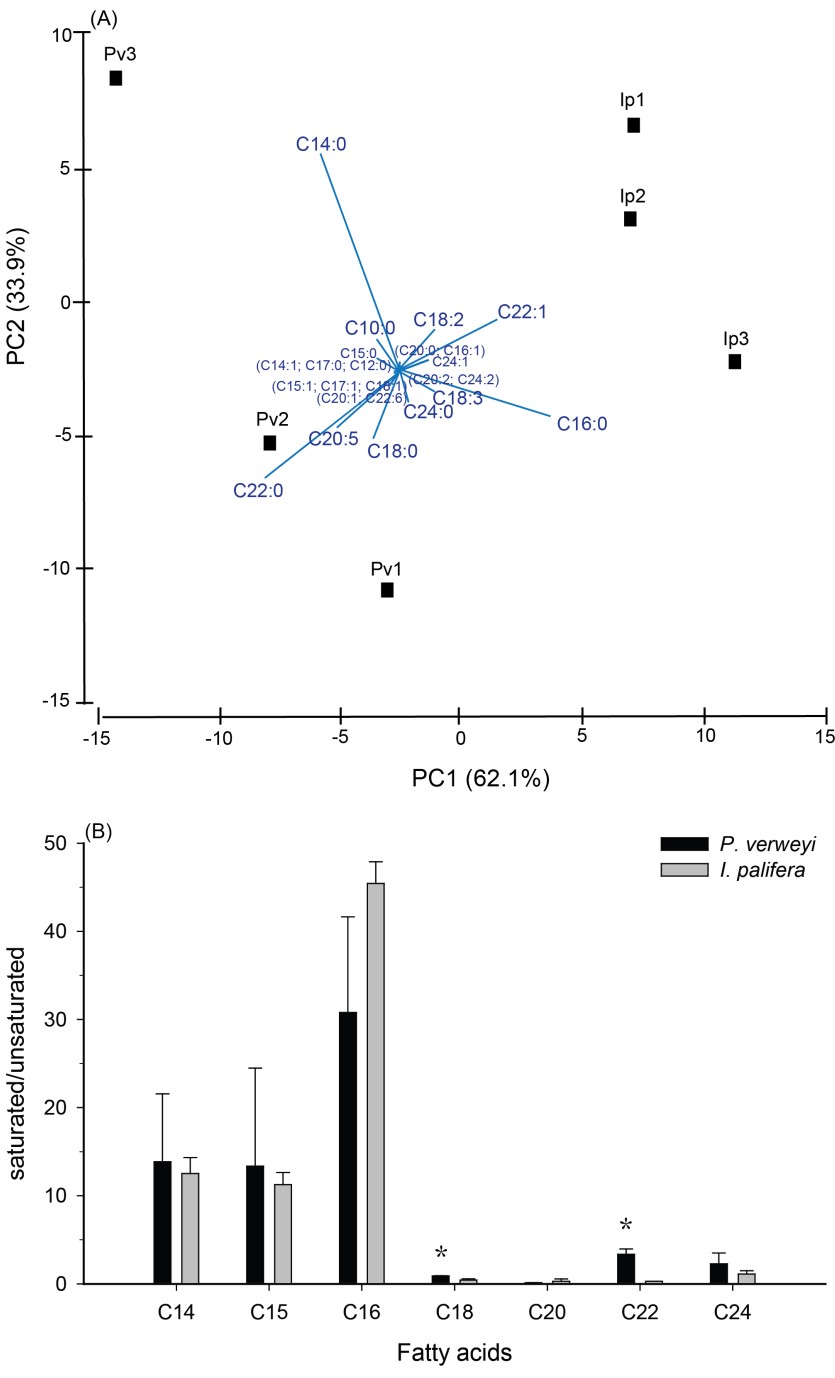

**Figure 2** **Comparison of fatty acid composition between menthol-bleached *Platygyra verweyi* and *Isopora palifera*.** Principal component analysis of fatty acid composition (A) and degree of saturation of each dominant fatty acid (B) are presented. Stars above means are significantly different at $P = 0.05$ (*t*-test).

indicated *I. palifera* contained a higher percentage of a short chain FA, palmitic acid, and a long chain unsaturated FA, erucic acid (22:1). The PC2, accounting for 33.9% of the variance, was highly correlated with the relative percentage of myristic acid in the coral host. When comparing FA in groups, *P. verweyi* displayed significantly higher saturated ($P < 0.05$) and lower mono-unsaturated FA ($P < 0.01$) than *I. palifera*, but the two species showed no significant differences in poly-unsaturated FA (Table S1). Further, to examine the degree of saturation of each FA longer than C14 indicated (Fig. 2B), both coral species displayed comparable and high saturation levels (saturated/unsaturated: 10~40) in short chain FA (C14, C15, and C16), but low and varied saturation levels (0.3~3.4) in long chain FA (C18, C20, C22, and C24). Moreover, the degrees of saturation in C18 and C22 of *P. verweyi* FA were significantly higher than in *I. palifera* ($P < 0.05$), and in C22 of *P. verweyi* were about 12-fold higher than in *I. palifera*.

The activity of antioxidant enzymes SOD and CAT in the heat-treated corals were also examined. Given that part of the *I. palifera* samples were completely bleached after 24 h at 31 °C, the determination of enzyme activity in this species was conducted only with the samples under 24 h incubation. SOD activity in both coral species indicated that enzyme activity varied between incubation times ($P < 0.05$). As indicated in Fig. 3A, *P. verweyi* had stable SOD activity at about 100 U mg$^{-1}$ within 48 h at 31 °C, and then significantly increased to $181 \pm 51$ U mg$^{-1}$ at 72 h of treatment ($P < 0.05$). *I. palifera* contained <10% SOD of that of *P. verweyi* ($P < 0.01$). In response to heat treatment, both coral species significantly doubled their SOD activity at different time regimes ($P < 0.05$), which occurred at 24 h and 72 h of heat treatment for *I. palifera* and *P. verweyi*, respectively. While CAT analysis indicated that enzyme activity in both coral species did not significantly vary between incubation times ($P > 0.05$), *P. verweyi* displayed nearly 3 × higher CAT activity than *I. palifera* ($P < 0.01$) (Fig. 3B).

## Composition of the bacterial community

Bacterial composition in *P. verweyi* and *I. palifera* before and after 24 h of heating at 31 °C was also analyzed. The non-chimera reads of all but one of the samples ranged from 13,698 to 78,839 (37,257 on average). Only one sample from heat-treated colony 1 of *I. palifera* showed a low read of 2,909. Excluded chimera reads in all samples were < 5% (2.4% on average). Total identified OTUs were 2,037, in which 1,955 OTUs were classified as bacteria. Composition analysis indicated that both species before and after 24 h heating were dominated (>85%) by Proteobacteria (Fig. S3). At the class level, both coral species were consistently dominated by γ- and α-proteobacteria, but before-heated *P. verweyi* contained significantly more γ-proteobacteria ($84.4 \pm 1.7\%$) and less α-proteobacteria ($10.0 \pm 1.7\%$) than the corresponding *I. palifera* ($56.2 \pm 13.8\%$ γ-proteobacteria and $28.9 \pm 5.9\%$ α-proteobacteria; $P < 0.05$; Fig. S4). Heating at 31 °C for 24 h did not change the dominance of γ- and α-proteobacteria in both species ($P > 0.05$; Fig. S4). When comparing γ-proteobacteria composition between the two coral species (Fig. 4), the MDS plot indicated that, before and after heating, bacterial composition in *P. verweyi* was completely different from *I. palifera* (ANOSIM, $P = 0.002$). Also, no significant changes

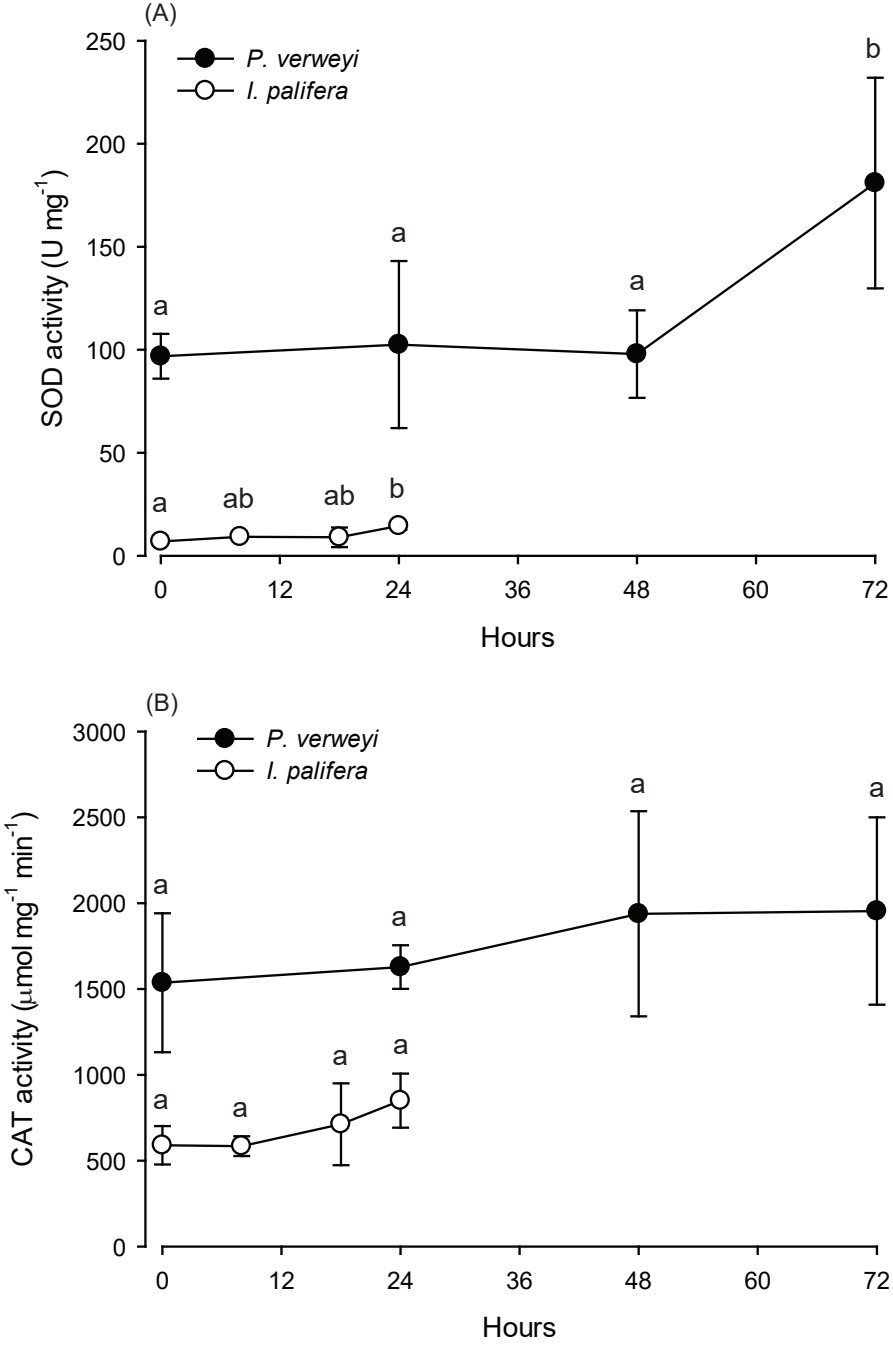

**Figure 3** Changes in the antioxidant activity of *Platygyra verweyi* and *Isopora palifera* host tissues when treated at 31 °C. The data points in (A) are the activity of superoxide dismutase (SOD) and in (B) are catalase (CAT). Means followed by the same letter are not significantly different at $P = 0.05$ (Tukey's *post hoc* analysis).

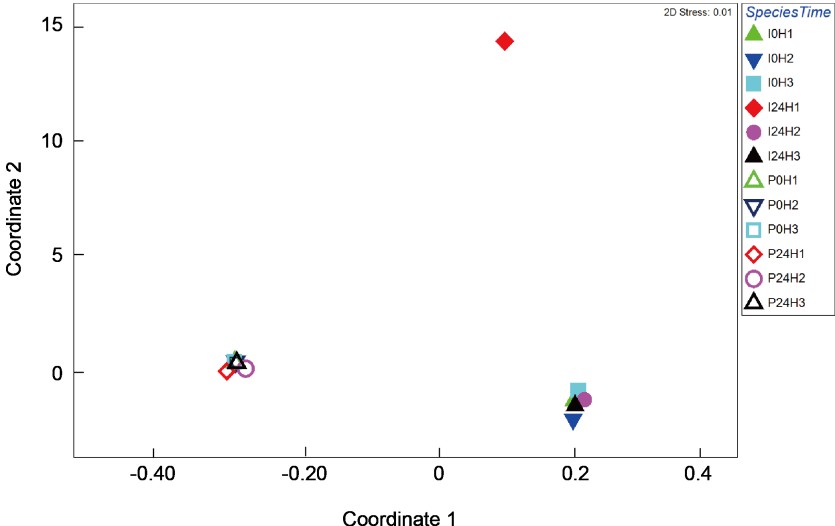

**Figure 4** **Multi-dimensional scaling (MDS) ordination of arc cosine-transformed relative abundance (%) in γ-proteobacteria composition associating with *Platygyra verweyi* and *Isopora palifera* (stress = 0.01).** Empty symbols represent data from *P. verweyi* at 31 °C for 0 h (P0H1-3) and 24 h (P24H1-3); and filled symbols represent that from *I. palifera* at 31 °C for 0 h (I0H1-3) and 24 h (I24H1-3).

in γ-proteobacteria composition were found in corals with 24 h heat treatment (Fig. 4), except for the colony 1 of heat-treated *I. palifera*.

More than 50% (59.7 ± 2.9% for before-heating and 62.7 ± 14.3% for after-heating samples) of the genera of γ-proteobacteria in *P. verweyi* were indicated (Fig. 5) to be *Endozoicomonas*, but the latter's relative abundance in *I. palifera* was low (<2.0% for before-heating and <0.1% for after-heating). Moreover, heating for 24 h did not change the relative abundance of *Endozoicomonas* in *P. verweyi*. The most dominant γ-proteobacteria in *I. palifera* was γ-proteobacteria/unclassified, which, however, varied among colony replicates during heat treatment. The relative abundance of γ-proteobacteria/unclassified of colonies 1 and 2 decreased from 21.7~32.6% to 0.7~8.2% after heating, but colony 3 displayed a slight increase from 14.5% to 19.0%. A huge increase in the relative abundance of *Vibrio* was found in colony 1 of *I. palifera* after heating, increasing from 1.0 to 64.7%, while the relative abundance of *Vibrio* in heated colonies 2 and 3 remained at low levels, increasing only from 0.2~0.7% to 0.6~3.6%. Note that unusually high relative-abundance of *Vibrio*, low non-chimera reads and manifesting as an outgroup in the MDS analysis were all came from the heat treated colony 1 of *I. palifera*.

## Thermal tolerance of corals with the manually altered Endozoicomonas association

When *P. verweyi* was treated with antibiotics for nine days, only the replicates from colony 2 at 31 °C/48 h showed no *Endozoicomonas* detection (Fig. S5B); however, total bacterial counts in their coral tissue extracts were reduced by more than 80%. The attempted inoculation of *E. montiporae* in *I. palifera* resulted in no positive infection in most colony replicates, except those from colony 2 at 31 °C/24 h, /48 h, and /72 h (Fig. S5C). When

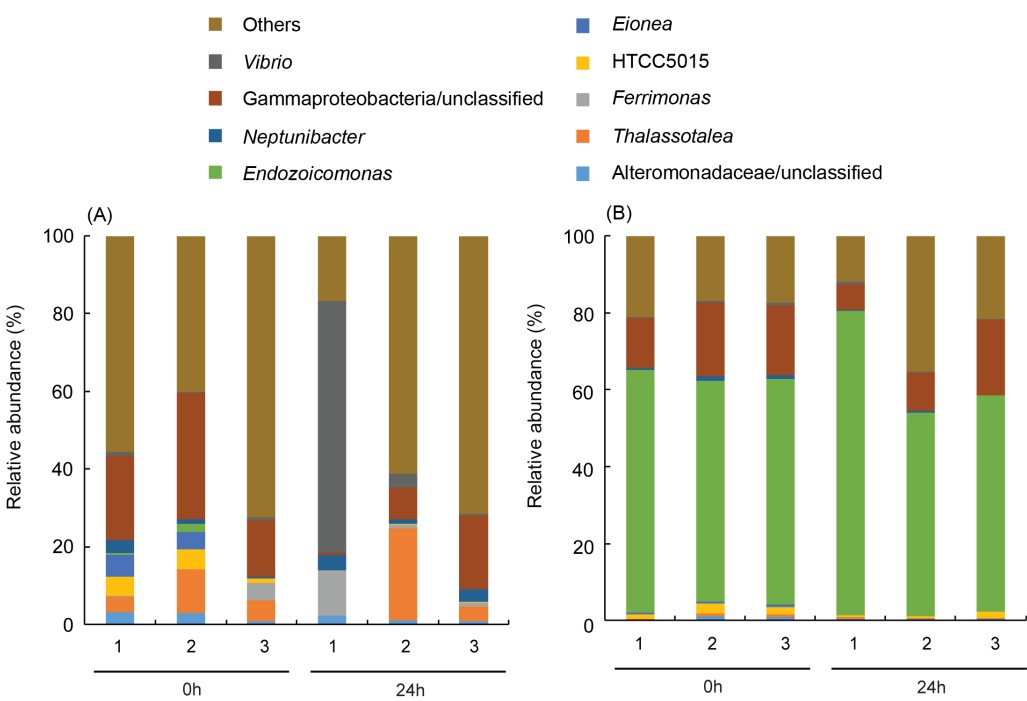

**Figure 5 Dominant bacterial genus composition in the γ-proteobacteria class in the corals.** The data in (A) are derived from *Isopora palifera* and (B) from *Platygyra verweyi*, and the numbers (1, 2, 3) represent three colony replicates. Genera with ¡ 1% abundances were grouped into others.

subjecting treated corals to 31 °C (Table 1) both the antibiotic-treated *P. verweyi* and those recovering from antibiotic treatment displayed no significant changes in algal density after 72 h incubation, compared with pre-treated and 25 °C treated controls ($P > 0.05$). Table 1 also indicates that, with or without *E. montiporae* infection, *I. palifera* displayed a significant decrease in algal density from $(2.3 \pm 1.2) \times 10^6$ cm$^{-2}$ to $(6.7 \pm 2.6) \times 10^4$ cm$^{-2}$ for *Endozoicomonas* infection and $(6.6 \pm 8.6) \times 10^5$ cm$^{-2}$ for no *E. montiporae* infection when incubated at 31 °C for 72 h.

# DISCUSSION

This study indicates that coral hosts might require extra high antioxidant enzyme activity to counteract acute thermal stress, even when already associated with thermal resistant algal symbionts. In Kenting, southern Taiwan, a nuclear power plant along the western side of Nanwan Bay has continuously discharged hot water into ambient seawater since 1984. Part of the heated seawater is trapped in a bay on the west side of the nuclear power plant outlet (NPP-OL) because of a near-shore current and tides (*Chiou, Cheng & Ou, 1993*), which elevate average seawater temperatures 2.0∼3.0 °C higher than at other coral reef sites in Kenting (*Fan, 1991*; *Pier, 2011*), also see Keshavmurthy et al. 2012). The hot water discharged in this area has impacted the marine ecology within the area (*Chiou, Cheng & Ou, 1993*; *Hung, Huang & Shao, 1998*; *Jan et al., 2001*; *Hwang, Tsai & Lee, 2004*). After more than 30 years of regular hot water influx and exposure to several

**Table 1 Changes in the algal density of *Platygyra verweyi* and *Isopora palifera* with and without manipulating the *Endozoicomonas* association.** The endogenous *Endozoicomonas* spp. in *P. verweyi* were inhibited with antibiotics (+Ab) for 9 days and then recovered in clean seawater (recovery from +Ab) for 1 week. *I. palifera* were inoculated with *E. montiporae* (+Endo.) or a seawater blank (-Endo.) for 2 h per day and repeated for 7 days. Algal density ($\times 10^6$ cells cm$^{-2}$) in the colonies of *P. verweyi* before antibiotics treatment and *I. palifera* before *Endozoicomonas* infection were $(2.6 \pm 0.8) \times 10^6$ cm$^{-2}$ and $(2.3 \pm 1.2) \times 10^6$ cm$^{-2}$, respectively. Data are expressed as relative percentage of algal density in each treatment to that in the pre-treated colony.

| Hours at 31 °C | *P. verweyi* | | *I. palifera* | |
|---|---|---|---|---|
| | +Ab | recovery from +Ab | +Endo. | -Endo. |
| | | (%) | | |
| 0 | $140 \pm 47$ | $130 \pm 53$ | $87 \pm 26$ | $90 \pm 42$ |
| 24 | $118 \pm 18$ | $140 \pm 26$ | $26 \pm 15$ | $33 \pm 14$ |
| 48 | $188 \pm 11$ | $111 \pm 57$ | $19 \pm 17$ | $30 \pm 16$ |
| 72 | $153 \pm 72$ | $128 \pm 30$ | $4 \pm 4$ | $21 \pm 21$ |
| 72 h at 25 °C | $115 \pm 14$ | $123 \pm 22$ | $80 \pm 26$ | $69 \pm 11$ |

instances of heatwaves, the coral community at NPP-OL has dramatically changed and lost almost all branch coral species from its shallow waters (*Keshavmurthy et al., 2014*). The remaining "thermal adapted" coral species in those shallows display constant associations with thermal resistant algal genotypes such as *D. trenchii* or *Cladocopium* sp. (C15) (*Keshavmurthy et al., 2014*). Unexpectedly, this study found that two "thermal adapted" and *D. trenchii*-associating coral species, *P. verweyi* and *I. palifera*, display distinctly higher bleaching resistant capability in former species than the latter one when confronted with acute thermal stress (Fig. 1B). The performance of *Isopora* on thermal bleaching might be different from that in Fig. 1B, if treated with slowly heating. However, even treated with a sudden rising of seawater temperature to 31 °C, *Platygyra* displayed no algal loss (Fig. 1B). This finding indicated *Platygyra* might have certain thermal resistant characteristics, which not found in *Isopora*. Due to both corals harboring the same symbiont strain, this phenomenon might not be derived from the performance of algal symbiont. High maximum quantum yield in bleaching *Isopora* after 32 h of heat treatment (Fig. 1A) partially supported this statement. Therefore, this study focused on characterizing the differences in coral hosts.

Coral tissue thickness indicates that both coral species belong to the thick-tissue group, consistent with other coral species prone to higher survivorship after thermal bleaching as indicated by *Loya et al. (2001)*. The respiration response to temperature (RBT) also indicates that both species might have similar acclimatization temperatures, which is consistent with their habitat environment (*Wang et al., 2019*). However, none of these explanations explain the distinctly different performance in acute thermal response between the two species. The differences between *P. verweyi* and *I. palifera* hosts showed in FA composition, antioxidant enzymes, and microbiome composition, any or all of which could directly or indirectly account for their distinctly different bleaching responses to acute thermal stress.

The FA analyses of menthol-bleached *P. verweyi* and *I. palifera* displayed two separate compositions, both of which were lacking in typical alga-derived polyunsaturated fatty acids (*i.e.,* C18:4, C22:5, and C22:6), which are markers of algal symbiosis (*Papina, Meziane & Van Woesik, 2003*), indicating that their FA compositions were solely derived from the coral hosts. Consistent with the more thermal tolerant response in *P. verweyi*, this coral displayed significantly higher saturated and lower mono-unsaturated FA than *I. palifera*. Moreover, the FA composition of *P. verweyi* was also dominated (14.0 ± 1.5%) by a long-chain saturated FA (C22:0), in which it was 4 × more abundant than in *I. palifera*. FA became more saturated and contained longer carbon chains in thermal tolerant *P. verweyi*, which coincided with remodeling of membrane lipids at higher temperatures by adjusting FA composition to compensate for membrane fluidity. This is a typical response of a membrane to rising temperature, known as homeoviscous adaptation (HVS) (*Sinensky, 1974*). A similar adjustment in FA composition in response to temperature was also found in Symbiodiniaceae algae (*Tchernov et al., 2004*) and the giant clam (*Dubousquet et al., 2016*). The unusually high abundance of behenic acid (C22:0) in *P. verweyi* is worthy of further study to explore the effect of long chain saturated FA on coral membrane stability and its contribution to mitigate coral bleaching under heat stress.

To investigate the role of the coral host on reducing coral bleaching stress, *Baird et al. (2009)* summarized five potential mechanisms, including constitutive photoprotective components, antioxidant systems, heat shock proteins, and shifting from heterotrophic feeding to gain energy. Except for adjusting the energy source, all efforts by coral hosts to alleviate thermal bleaching come down to reducing oxidation stress. In this study, we did not examine the production of heat shock protein in both species. However, *P. verweyi* was found to contain several-fold higher antioxidant enzyme activity (SOD and CAT) than *I. palifera* before and during heat treatment. Especially, enzyme activity before heat treatment had about 14 times higher SOD ($97 \pm 11$ Umg$^{-1}$) and 3 times higher CAT ($1{,}536 \pm 405$ $\mu$mol mg$^{-1}$min$^{-1}$) in *P. verweyi* than in *I. palifera* (SOD: $7 \pm 1$ Umg$^{-1}$ and CAT: $590 \pm 112$ $\mu$mol mg$^{-1}$min$^{-1}$). The function of SOD in the anti-oxidation process is to convert reactive oxygen species superoxide ($O_2^{-1}$) to $H_2O_2$, followed by decomposing to $H_2O$ and $O_2$ by CAT (*Lesser, 2006*). Extra high constitutive SOD in *P. verweyi* suggests that these corals, after harboring thermal-tolerant algal symbionts adapted to the warmer habitat at NPP-OL, had also built up an efficient primary defense system against temperature-induced oxidative stress. The high SOD activity in *P. verweyi* from NPP-OL was obviously stimulated by high and fluctuating temperatures, because the same coral species from a nearby and temperature stable location, Wanlitong, contained levels of SOD ($11 \pm 3$ Umg$^{-1}$, JT Wang, 2020, personal observations) that were comparable to those of *I. palifera* in this study. Evidence of the adaption of *P. verweyi* to high and fluctuating NPP-OL was also found in this coral's key respiration enzyme, malate dehydrogenase (*Wang et al., 2019*). In contrast to *P. verweyi*, *I. palifera* seemed to only replace its temperature sensitive algal symbiont with a more temperature resistant one. Further study is needed to determine whether it is a faster evolution rate or higher phenotypic plasticity that confers upon *P. verweyi* more adaptability to higher and fluctuating temperature habitats.

*P. verweyi* also displayed a consistent association with the coral symbiotic microorganism *Endozoicomonas* at a relatively high (>1/3) abundance of total bacterial population; this was not found in *I. palifera*. However, depleting *Endozoicomonas* from *P. verweyi* with antibiotics did not abolish the coral's capability to resist bleaching at 31 °C, and also did not change ROS and CAT activities significantly. Because >80% of the bacterial counts of *P. verweyi* had been depleted with *Endozoicomonas*- sensitive antibiotics (*Yang et al., 2010*; *Sheu et al., 2017*; *Chen, Lin & Sheu, 2019*), corals retaining their bleaching-resistance capability at 31 °C likely was not due to invalid antibiotic treatment. *Endozoicomonas*- positive detection by PCR examination (Fig. S5A) might be derived from DNA residues of the bacterium in coral tissue. Directly inoculating cultured *E. montiporae* to *I. palifera* was barely successful. However, even when successfully infected with *E. montiporae*, *I. palifera* still displayed significant bleaching after 24 h at 31 °C. Besides *E. montiporae*, infection tests were also conducted with *E. acroporae* (*Sheu et al., 2017*) and *E. coralli* (*Chen, Lin & Sheu, 2019*), with the same results as those from *E. montiporae* (JT Wang, 2020, personal observations). The failure to infect *I. palifera* with *Endozoicomonas* might be attributed to host specificity (*Ding et al., 2016*; *Neave et al., 2017*), but high variability in seasonal microbial partnerships in *I. palifera* (*Chen et al., 2011*) also suggests that these corals might not be able to establish stable associations with certain bacterial strains as do other corals (*Neave et al., 2017*; *Pogoreutz et al., 2018*).

*Endozoicomonas* are generally assumed to be essential in coral holobiont functioning due to their widespread prevalence and high abundance in many coral species (*Bayer et al., 2013*; *Jessen et al., 2013*; *Meyer et al., 2016*; *Glasl, Herndl & Frade, 2016*; *Neave et al., 2016*; *Gignoux-Wolfsohn, Aronson & Vollmer, 2017*; *Neave et al., 2017*) and apparent metabolic versatility (*Yang et al., 2010*; *Hyun et al., 2014*; *Ding et al., 2016*; *Neave et al., 2017*). Reductions in the abundance of *Endozoicomonas* in stressed, diseased, or bleached corals were also reported in a number of studies, suggesting that the pervasive abundance of *Endozoicomonas* might be closely linked to a healthy status in corals (*Bourne et al., 2008*; *Cárdenas et al., 2012*; *Gignoux-Wolfsohn, Aronson & Vollmer, 2017*; *Meyer et al., 2016*; *Morrow et al., 2014*; *Roder et al., 2015*; *Röthig et al., 2016*; *Ziegler et al., 2016*). However, some coral species seem to form more obligate associations with *Endozoicomonas*, even under different stresses (*Sharp, Pratte & AH, 2017*; *Pogoreutz et al., 2018*; *Shiu et al., 2020*). *P. verweyi*, as revealed in this study, also displayed a constant association with *Endozoicomonas*; however, a reduction in *Endozoicomonas* abundance did not impair symbiosis between Symbiodiniaceae algae and the coral even when heated at 31 °C. *Endozoicomonas*, unlike Symbiodiniaceae algae residing in coral cells, forms cell aggregates in coral tissues (*Bayer et al., 2013*). Therefore, it is reasonable to expect that Symbiodiniaceae algae and *Endozoicomonas* may undergo different evolutionary tracks to form symbioses with coral, and may have separate interactions with their host even though both symbionts are altogether beneficial to the host.

## CONCLUSION

In summary, rising temperatures drove coral species at NPP-OL to associate with thermal-resistant symbiont strains, but the modification in coral hosts for adapting to the high

temperature environment was to diversify between species. Our results imply that bleaching resistance might be upgraded by having more-saturated FA in coral membrane lipids and especially extra high coral SOD activity, which would benefit future attempts on assisted evolution in coral restoration. Coral symbiotic bacteria like *Endozoicomonas* are believed to play important roles on coral survivorship under stress, but their contributions might be indirect.

# ACKNOWLEDGEMENTS

The authors would like to thank members of the Coral Reef Evolutionary Ecology and Genetics (CREEG) Group, Biodiversity Research Center, Academia Sinica (BRCAS), for field support, Prof. Youk-Meng Choong for fatty acid analysis, Prof. Win-Ming Chen for providing *Endozoicomonas* culture, and Buford Pruitt for English editing.

## Funding

This work was supported by the Ministry of Science and Technology Taiwan (MOST 107-2621-B-127-001 and MOST 108-2621-B-127-001) to JTW. The funders had no role in study design, data collection and analysis, decision to publish, or preparation of the manuscript.

## Grant Disclosures

The following grant information was disclosed by the authors:
The Ministry of Science and Technology Taiwan: MOST 107-2621-B-127-001, MOST 108-2621-B-127-001.

## Competing Interests

The authors declare there are no competing interests.

## Author Contributions

- Jih-Terng Wang conceived and designed the experiments, prepared figures and/or tables, authored or reviewed drafts of the paper, and approved the final draft.
- Yi-Ting Wang and Pei-Wen Chiang performed the experiments, prepared figures and/or tables, and approved the final draft.
- Chaolun Allen Chen, Pei-Jei Meng and Sen-Lin Tang conceived and designed the experiments, authored or reviewed drafts of the paper, and approved the final draft.
- Kwee Siong Tew analyzed the data, prepared figures and/or tables, and approved the final draft.

## Field Study Permissions

The following information was supplied relating to field study approvals (i.e., approving body and any reference numbers):

Pingtung County Government and National Kenting National Park approved this research.

## Data Availability

The data is available at the SRA: PRJNA736837.

## Supplemental Information

Supplemental information for this article can be found online at http://dx.doi.org/10.7717/peerj.12746#supplemental-information.

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
