# Peer review of "Extra high superoxide dismutase in host tissue is associated with improving bleaching resistance in “thermal adapted” and Durusdinium trenchii-associating coral"

_PeerJ, doi:10.7717/peerj.12746_

## Round 0.1 · original submission · Major Revisions

I have read your manuscript and have not sent it out for review, because, in my opinion, the English language is not of high enough standard. I kindly ask for you to find a native English speaker to revise the document or to hire an English proofreading service to do so. Once you have done this I will be more than happy to send your manuscript for peer review.

---

## Round 0.2 · Major Revisions

Two expert reviewers have evaluated your manuscript and their comments can be seen below. This study is interesting and presents a complete story. However, there are are number of issues that need to be taken into account in a revised version of the manuscript. Please ensure that you address all of the comments in a rebuttal.

Reviewer 1 ·

Basic reporting

Paper title: Extra high superoxide dismutase in host tissue improves bleaching resistance in “thermal adapted” and Durusdinium trenchi-associating coral.

The present manuscript presents interesting results that explains one of the potential tolerance mechanisms of metaorganism to ecological stressors. They have utilized biochemical approach to quantify the ROS protective enzymes and fatty acid composition of thermally adapted corals that might aid them in avoiding bleaching. There were some interesting findings like higher SOD, and catalase activity and higher levels of saturated long-chain fatty acid composition in thermally acclimated and temperature resistant coral P. verweyi. A noteworthy finding in this work is the contribution of the host factors in providing protection from bleaching, as both the higher temperature acclimated corals contained the same thermal resistant algae D. trenchi, but possessing D. trenchi did provide a considerable resistance compared to the thermal sensitive Stylophora pistillata. The authors also attempted to determine the role of bacteria, specifically Endozoicomonas sp., however they could not detect any clear and detectable role of bacteria in providing bacterial bleaching resistance.

There was a clear improvement in the manuscript language and grammar compared to the previous version. The manuscript is also well structured with clearly defined objectives, results and conclusions. However, I would like to present some points for corrections and some questions that need clarifications + some suggestions.

Experimental design

The experimental design was simple yet elegant, where they compared 2 thermal adapted corals and used a thermal sensitive coral as an additional control. The samples were utilized for estimating physio-chemical and microbiome parameters.

Validity of the findings

Here are some suggestions for minor revision.
• The authors comment that high SOD and CAT activities were responsible providing protection from bleaching. However, this is a strong statement to make because enough conclusive functional experiments were missing like using SOD or CAT inhibitors or altering the activity of these enzymes in any other way and then checking the effect of temperature on bleaching. Hence I would also suggest to change the title of the paper and modify it to “….in host tissue is associated with improving bleaching resistance in…..”.
• Continuing with the previous point, the authors did not measure the ROS levels in the thermal acclimated corals. Are ROS levels different in both the thermal acclimated corals and do they increase after providing heat stress and are the levels comparable in both species? Can authors provide some clarifications here?
• The authors have used Principal Component Analysis (PCA), but they have referred to it as Primary component analysis at multiple places. It needs to be rectified.
• Fig. 2A represents the PCA plot for the fatty acid composition. Authors mention that fatty acids C20:5 and C22:0 mainly accounted for the observed variation (lines 299-300), however, it is clear from the plot in fig.2A that FA C18:00 is also playing a role in the observed variation. Can authors clarify this point and if it fits, include C18:00 also in the discussion? Please indicate the percent variation of each axis on the plot itself on the respective axes.
• The authors mention that sample 1 of I. palifera had low reads and hence it was excluded from analysis for MDS plot. However, the same sample is discussed later to be possessing a high abundance of Vibrio compared to other samples of the same treatment group. I would suggest not to make any conclusions or suggestions altogether from this sample or else include it for all the analysis. How reliable is the data from this sample to substantiate the high Vibrio abundance?
• Authors have used E. montiporae for colonizing I. palifrea but the colonization failed for most colonies. Could it because of the difference between the species of Endozoicomonas existing on the local coral and the one that is under cultivation? Did the authors attempted to cultivate the natively associated Endozoicomonas?
• There are typos in the following lines: 246, 285, 312 (CA->CAT).
• Can authors comment, how might long chained saturated fatty acids contribute to provide resistance against bleaching?

Reviewer 2 ·

Basic reporting

Wang et al. analyzed two coral genera maintaining the same stress tolerant Symbiodiniaceae species, provided more experimental evidence, further supporting antioxidant enzymes as well as saturated fatty acid contributing to coral heat tolerance. Despite the short experimental period, the whole story is logical and complete. My major concern is about the microbial community analysis part. Large part of sequences in Isopora were unassigned at genus level, it looks unusual to me. Thus, it’d be better to try some other methods for the taxonomy assignment, e.g. RDP, and it may provide more information and different following results. Besides, although I’m also not a native speaker, the writing still needs further improvement. Some parts cause much confusion, such as pre heat, it mostly means heat something beforehand.

Some other comments
Line 217-220 The accession numbers of raw sequence reads as well as analysis code shall be provided here.

Line 272-275 Statistical analysis between the coral species is needed to provide more solid evidence.

Line 318-323 Although the comparisons between coral species have been made like “<10%SOD”, “3x higher”, it’d be better to also provide statistical results here.

Figure 2(A) The labels can be barely read. Needs to find a better way to display/sperate them.

Figure 4 The colors of the symbol cause much confusion. Need to find a better way to display.

Figure 5 Genus name shall be in italic. Besides, according to the statements, it’s Platygyra containing more Endozoicomonas, but in this figure, samples in B (Isopora) appear to have more Endozoicomonas.

Experimental design

no comment

Validity of the findings

Discussion part: because the experimental period in this study is quite short, and also just gave corals acute heat stress, but in natural environment, we’d rather see gradually increasing temperature, which Isopora with durusdinium trenchii is able to acclimate possibly, not to mention Symbiodiniaceae usually exhibit higher antioxidant enzyme activity than coral hosts. And it’s also possible Isopora can gradually acclimate to the high temperature if the experiment lasts longer. These shall all be considered in the discussion part.

Additional comments

Supporting material
Bacteria classification This excel file is too difficult to read. It looks like a draft to me. Need to label clearly the meanings of these numbers.

---

## Round 0.3 · Major Revisions

An expert reviewer has provided feedback on your revised manuscript. Please ensure you respond to each comment or suggestion in a revised version of the manuscript.

Reviewer 2 ·

Basic reporting

The authors have addressed some of my comments. But there still remain some unsolved ones.

I still insist authors give RDP another try for taxonomy assignment. To my experience, RDP usually gives less than 10% unclassified sequences for coral microbial samples. While in the present study, some samples got 20-30% unassigned sequences at genus level. The taxonomy assignment will completely affect the following analysis and explanation. And also, RDP is a widely accepted classifier tool.

“Comment: The accession numbers of raw sequence reads as well as analysis code
shall be provided here (Line 217-220).
Response: Accession number to raw sequence data was added (L232-233).”
The analysis code/script was still not provided here.

“Comment: Line 272-275 Statistical analysis between the coral species is needed
to provide more solid evidence.
Response: P values were added to show statistically differences between species.”
I don’t find statistical analysis added up here. It’s for the algal density part.

“Comment: Line 318-323 Although the comparisons between coral species have
been made like “<10%SOD”, “3x higher”, it’d be better to also provide
statistical results here.
Response: Statistical results were added.”
I’m confused about the analysis here. The p-value represents the significant difference or “10%SOD/3xhigher”? Please clarify.

“Comment: Figure 2(A) The labels can be barely read. Needs to find a better way
to display/sperate them.
Response: The barely read labels in Figure 2(A) were because these FA were not major components to distinguish Isopora from Platygyra. In order to highlight the difference, we focused on the FA with high eigenvectors.”
I understand some may not be vital components. If so, just remove these uncrucial ones or if you do want to display them, it’s necessary to be presented very clearly.

“Comment: Bacteria classification This excel file is too difficult to read. It looks
like a draft to me. Need to label clearly the meanings of these numbers.
Response: Bacterial classification table has been tidied up.”
The table still needs more improvement. For results from calculation, needs to keep the same digits (e.g. 2/4) after decimal points. After if the numbers represent percentage, the “%” should be added.

Experimental design

NA

Validity of the findings

NA

Additional comments

NA

---

## Round 0.4 · Minor Revisions

Two reviewers have evaluated your revised manuscript and have suggested some minor changes.

Reviewer 1 ·

Basic reporting

The authors have carefully and satisfactorily addressed all the comments from the previous round of review. Just two small comments:
> Please correct the typo in the line 467-468: …might not be able to establish stable…..
> Please provide the scale/co-ordinates of the X and Y axis with labels on the MDS plot of the figure 4.

Experimental design

no comments

Validity of the findings

no comments

Additional comments

Just make the 2 changes mentioned in the 'Basic reporting' section.

Reviewer 2 ·

Basic reporting

The authors still didn't provide their running scripts (not just accession number) here. But considering the analysis used in the study is common, it's not that a big deal if the codes are missed. Still it'd be better if the authors can provide as supplmentary material during their final check.

Experimental design

NA

Validity of the findings

NA

Additional comments

NA

---

## Round 0.5 · accepted · Accept

I am satisfied with the changes made to the manuscript.